# Advances in Tissue Engineering for Disc Repair

**Chang Kyu Lee** [1,†] [ID], **Dong Hwa Heo** [2,†] [ID], **Hungtae Chung** [2], **Eun Ji Roh** [3], **Anjani Darai** [3] [ID], **Jae Won Kyung** [3] [ID], **Hyemin Choi** [3], **Su Yeon Kwon** [3], **Basanta Bhujel** [3] and **Inbo Han** [3,*] [ID]

1    Department of Neurosurgery, Keimyung University School of Medicine, Dongsan Medical Center, Daegu 42601, Korea; faslcklee@gmail.com
2    Department of Neurosurgery, Seoul Bumin Hospital, Seoul 07590, Korea; spinesurgery@naver.com (D.H.H.); bregma@bumin.co.kr (H.C.)
3    Department of Neurosurgery, CHA University School of Medicine, CHA Bungdang Medical Center, Seongnam-si 13497, Korea; morolro@naver.com (E.J.R.); anjanianji09@gmail.com (A.D.); kyungjaewon88@gmail.com (J.W.K.); littlechoi88@gmail.com (H.C.); syunkwon@naver.com (S.Y.K.); basantabhujel86@gmail.com (B.B.)
\*    Correspondence: hanib@cha.ac.kr; Tel.: +82-31-780-1924
†    These authors contributed equally to this work.

**Abstract:** Intervertebral disc (IVD) degeneration is a leading cause of chronic low back pain (LBP) that results in serious disability and significant economic burden. IVD degeneration alters the disc structure and spine biomechanics, resulting in subsequent structural changes throughout the spine. Currently, treatments of chronic LBP due to IVD degeneration include conservative treatments, such as pain medication and physiotherapy, and surgical treatments, such as removal of herniated disc without or with spinal fusion. However, none of these treatments can completely restore a degenerated disc and its function. Thus, although the exact pathogenesis of disc degeneration remains unclear, there are studies examining the effectiveness of biological approaches, such as growth factor injection, gene therapy, and cell transplantation, in promoting IVD regeneration. Furthermore, tissue engineering using a combination of cell transplantation and biomaterials has emerged as a promising new approach for repair or restoration of degenerated discs. The main purpose of this review was to provide an overview of the current status of tissue engineering applications for IVD regenerative therapy by performing literature searches using PubMed. Significant advances in tissue engineering have opened the door to a new generation of regenerative therapies for the treatment of chronic discogenic LBP.

**Keywords:** intervertebral disc (IVD); regeneration; stem cell; biomaterials; scaffolds; tissue engineering

## 1. Introduction

Intervertebral disc (IVD) degeneration (IVDD) causes chronic low back pain (LBP), including discogenic back pain, significant health problems, and socioeconomic burden [1]. IVDD-induced discogenic LBP accounts for more than 40% of all LBP cases [2] and is considered as one of the top global causes of disability-adjusted life years [3].

IVDD is a process leading to loss of proteoglycans (PGs), destruction of the extracellular matrix (ECM), annular tears, development of disc herniation, and loss of disc height [4]. As a result of these anatomical changes, nerve root compression, spinal canal stenosis, and facet joint arthritis and hypertrophy can occur and can lead to chronic LBP and/or radiating leg pain with or without neurological deficits [5]. Moreover, the inflammatory environment of the degenerative discs and neurite sprouting have been suggested as the cause of discogenic LBP [6]. Thus, discogenic LBP is associated with complex interactions between the mechanical aspects of the IVD, inflammation, and the central or peripheral nervous system [7].

Chronic LBP due to IVDD may be considered for surgical treatments if there is no response to conservative treatments, such as medication and physical therapy. Surgical

treatments include discectomy to remove a herniated disc, spinal fusion surgery used to connect two vertebrae to limit the movement of the spinal motion segment, and artificial disc replacements designed to restore and maintain range of motion [8,9]. However, spinal fusion surgery does not restore the previous range of motion and mechanical load-bearing properties of the IVD. Moreover, spinal fusion can lead to adjacent segment disease, which is a typical long-term complication after spinal fusion surgery and further indicates disc herniation, spinal canal stenosis, or spondylosis at levels above or below the index fusion level [10]. Therefore, alternate biological therapies are needed prior to surgery to slow or reverse the progression of IVDD, which usually leads to pain and disability.

The mechanical properties of the IVD are critical for proper functioning. In vivo, the IVD is the load-bearing structure of the spine and is subjected to spinal tension, torsion, compression, and bending [11]. Anatomically, the normal IVD consists of the following three parts: (1) the nucleus pulposus (NP), which contains a highly hydrated gel-like matrix comprising PGs and type II collagen; (2) the annulus fibrosus (AF), which is composed of lamellae in which parallel type I collagen fibers located within each lamella are aligned and help the IVD maintain its integrity from bending, stretching, and twisting; and (3) the end plates (EP), which consist of osseous and two hyaline cartilages [12] (Figure 1). In recent years, the understanding of IVD development, cell biology, and mechanisms of IVD degeneration has significantly advanced, enabling the development of biological approaches for IVD regeneration [4]. Biological approaches include growth factor injection and gene and cell-based therapies, whereas tissue engineering approaches involve the restoration of the mechanical and biological properties of the tissue via the addition of biomaterials to the degenerated disc [13].

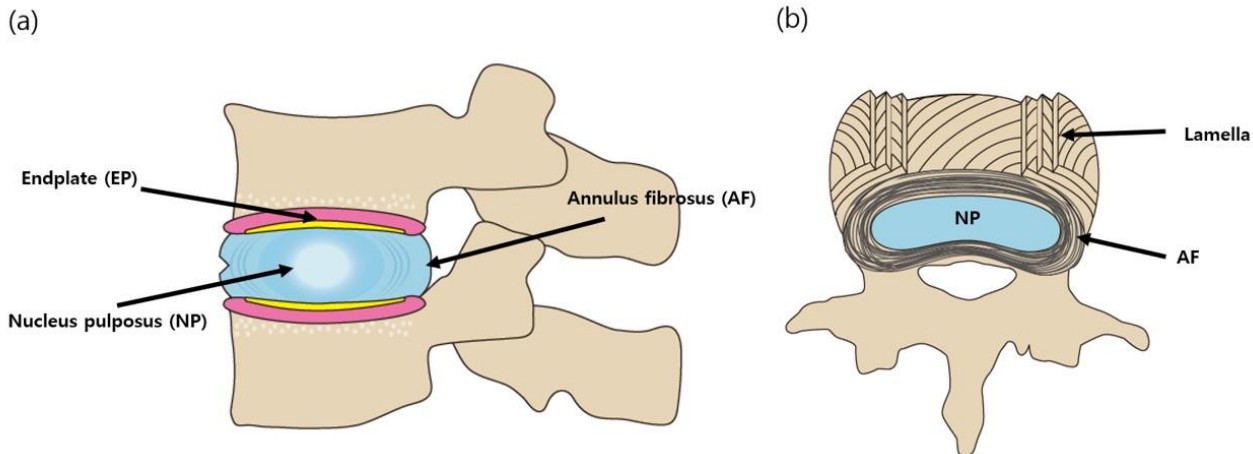

(a)  (b)

**Figure 1.** Schematic diagram of the intervertebral disc (IVD). (**a**) sagittal cross-section (**b**) axial cross-section.

Depending on the stage of IVDD, different treatment strategies for managing IVDD have been recommended [14]. Initial IVDD shows change in the NP and AF matrices, while IVDD induces progressive structural changes such as annular fissures, disc herniation, disc height reduction, and disc space collapse. The Pfirrmann Disc Grading is a useful scoring tool for evaluating IVDD on MRI T2-weighted images; Grade I: homogeneous disc with bright high intensity and normal disc height; Grade II: inhomogeneous disc but high intensity signal, clear distinction between NP and AF, and normal disc height; Grade III: inhomogeneous disc with an intermittent gray signal intensity, unclear distinction between NP and AF, and normal or slightly reduced disc height; Grade IV: inhomogeneous disc with low intensity dark gray signal intensity, no more distinction between the NP and AF, and slightly or moderately decreased disc height; Grade V: inhomogeneous disc with a hypointense black signal intensity, no more distinction between the nucleus and annulus, and collapsed disc space [15]. For severe nerve compression due to herniated disc and disc space collapse (Pfirrmann grades IV and V), surgical removal of herniated disc

and/or spinal fusion surgery are required. However, patients with discogenic LBP due to Pfirrmann grade II and III IVDD may receive regenerative molecular therapies such as growth factors, genes, and cell therapy with or without biomaterials. In addition, tissue engineered NP and AF could be applied for patients with Pfirrmann grades IV and V.

The aim of this review was to provide an overview of the current status of tissue engineering applications for the treatment of IVD regeneration. To obtain an overview of the current tissue engineering strategies utilized for the repair of degenerated discs, we conducted a literature search using PubMed (https://pubmed.ncbi.nlm.nih.gov/, from 1 January 2000 and 1 December 2020) and the following key words: "intervertebral disc", "regeneration", "stem cells", "biomaterials", and "tissue engineering".

## 2. Biological Approaches

### 2.1. Molecular Therapies

2.1.1. Growth Factors

In the early stages of IVDD, growth factor injection may rebalance the anabolic and catabolic pathways in the degenerative cascade [16]. Degenerated discs have been reported to be repaired by intradiscal injection of growth factors, including insulin-like growth factor-1(IGF-1), epidermal growth factor (EGF), basic fibroblast growth factor (bFGF or FGF2), transforming growth factor-β (TGF-β), bone morphogenetic protein-2 (BMP-2), BMP-7 (osteogenic protein-1; OP-1), and growth and differentiation factor-5 (GDF-5) by promoting cell proliferation and matrix synthesis in experimental models [17–19]. The mitogenic potential of human NP and AF cells can be stimulated by platelet-derived growth factor (PDGF), bFGF, and IGF-1 [20,21]. PDGF, a known angiogenic growth factor, has been shown to inhibit IVD cell apoptosis and promote anabolic gene expression [22]. In several animal experiments, intradiscal injection of BMP-7 has shown improvements in the disc height and NP proteoglycan content [23,24]. GDF-5, also known as BMP-14, is another anabolic protein that promotes cell proliferation and proteoglycan synthesis in degenerated discs [18]. Similar to BMP-7, many animal studies of intradiscal injection of GDF-5 have shown improved disc height, cell proliferation, and matrix synthesis [25,26]. Despite their efficacy, there are many concerns surrounding the clinical use of recombinant growth factors due to their short half-life, limited stability, high cost, and problems associated with binding large molecules to polymers [27]. Thus, the right carrier is a matter to consider. In recent years, the use of biodegradable microspheres for controlled local drug delivery has become a valuable approach to overcome the drawbacks of growth factors. Yan et al. demonstrated that injection of GDF-5 loaded into poly(lactic-co-glycolic acid) (PLGA) microspheres could improve regenerative efficacy of GDF-5 in a rat model [28].

2.1.2. Gene Therapy

A strategy to overcome the short half-life limitation of growth factors is to provide a sustained supply of growth factors within the IVD [29]. The therapeutic effect of gene therapy is based on the induction of target gene upregulation or downregulation. These genes are transferred using viral or non-viral vectors, which are either directly injected into the degenerated discs or transduced into cells [16]. Another strategy used for intradiscal gene therapy is gene expression downregulation, which is detrimental to the physiological balance of the disc and may, thus, lead to IVDD [16]. Hence, if gene therapy is performed properly, it can provide many benefits, including a more sustained target gene expression and long-term biological effects. Promising targets for gene therapy include both anabolic regulators, such as TGF-β, latent membrane protein 1(LMP-1), and SOX-9, and anticatabolic regulators, such as anti-ADAMTS (a disintegrin and metalloproteinase with thrombospondin motifs)-5, and TIMP (tissue inhibitor of metalloproteinases)-1 [13]. In terms of non-virus vector-mediated gene transfer, ultrasound targeted microbubble transfection method has been reported to improve the transfection efficiency of plasmid DNA in NP cells, and polyplex micelles made from a vector carrying miRNA-25-3p were used in an IVDD rat model [30]. Clustered Regulatory Interspaced Short Palindromic

Repeats-Associated Cas9 (CRISPR/Cas9) is an innovative technology that can be used to target other genes for IVDD treatment. Using the CRISPR/Cas9 gene-editing system in AF cells from patients with chronic LBP, knock out of the transient receptor potential vanilloid type 4 (TRPV4) gene induced the reduction in inflammation [30]. However, the use of gene therapy for IVD regeneration is currently limited to in vitro and in vivo animal studies due to safety concerns. There are no ongoing human clinical trials of gene therapies for IVD regeneration.

### 2.1.3. Summary

In order to overcome the shortcomings of molecular therapy, future studies will focus on the delivery and controlled release to the degenerated discs. In addition, the combination of growth factors, stem cells, and biomaterials should be a focus going forward.

### 2.2. Cell-Based Therapies

In the intermediate stages of IVDD, cell transplantation can be used to repopulate the disc. Although the NP cell phenotype has not been well defined, the adult NP contains cells similar to chondrocyte [31]. Cell transplantation to a moderately degenerated disc is a possible treatment that promotes disc regeneration by reproliferating cells that can restore the structural and functional properties of the degenerated disc or delay degeneration [27].

An optimal source of cells suitable for cell transplantation in the degenerative disc remains elusive. Several studies have demonstrated that implantation of IVD-derived cells delays the process of progressive degeneration and, in some cases, promotes disc regeneration in an animal model of IVDD [32,33]. However, since these cells are derived from normal NP tissue, they cannot be obtained without damaging the IVD [25].

Stem cells are a promising candidate cell source for use in cell-based therapies for IVD repair. Most of the stem cells used in disc regeneration experiments are derived from the bone marrow, adipose tissue, umbilical cord blood, umbilical cord Wharton's jelly, and synovium, because these cells are relatively easy to obtain and can differentiate into chondrogenic and IVD-cell lineages [34,35]. Implantation of MSCs into degenerated discs can prevent cellular apoptosis and inflammation (paracrine effect) or differentiate MSCs into NP cells to restore normal homeostasis and prevent or reverse further degeneration [13]. Injecting stem cells into the degenerated disc has been reported to increase the proteoglycan and water contents of the disc ECM (Table 1) [29].

Notochordal cells have been suggested to cause disc degeneration because their loss is associated with the onset of IVDD [33]. Sheyn et al. demonstrated that notochordal-like cells from human induced pluripotent stem cells (iPSCs) reduce IVDD in an injury-induced porcine model [33]. Therefore, the application of iPSCs is a hot topic in the field of IVD regeneration and has several advantages over embryonic stem cells (ESCs), such as fewer ethics and immune rejection issues [33]. However, iPSCs also have drawbacks in clinical applications, such as tumor formation by genomic integration of reprogramming factors [36].

For the clinical application of stem cells for IVD regeneration, MSCs treatment strategies, cell doses, and efficacy are being investigated in various experimental settings of IVDD and clinical trials [37]. The use of MSCs is generally considered safe and effective in preventing IVDD, but the rate of osteophyte formation has been reported to be around 2.7% [37,38]. Osteophyte formation is believed to be the result of implanted MSC leakage. Therefore, the application of scaffolding materials, such as fibrin, hyaluronan, or atelocollagen, is strongly recommended to prevent cell leakage and reduce the risk of ectopic osteoblast differentiation of MSCs [39–50].

**Table 1.** Cell-based therapies for IVD regeneration.

| Author | Cell Line | Effect |
|---|---|---|
| 2019, Shi et al. [40] <br> 2019, Sheyn et al. [33] | Neonatal human dermal fibroblasts <br> Rabbit, in vivo <br> Notochordal cells from human iPSCs | Increased regeneration markers <br> Reduction of disc degeneration in a porcine model |
| 2018, Teixeria et al. [41] | Human BM-MSC <br> Bovine, ex-vivo | Promoted cell migration and increased inflammatory cytokine expression |
| 2018, Wang et al. [42] | Rat BM-MSC <br> Rat, in vivo | Hypoxic pre-treatment of BM-MSC with $CoCl_2$ enhanced migration, decreased apoptosis, increased disc height, MSC numbers in the NP and AF, and extracellular matrix production |
| 2017, Maidhof et al. [43] | Allogeneic rat BM-MSC <br> Rat, in vivo | Cell therapies administered at an early stage of injury or disease progression may have greater chances of mitigating matrix loss |
| 2017, Hang et al. [44] | Autologous canine BM-MSC <br> Canine, in vivo | PET was more reliable than MRI for quantifying implanted BM-MSC survival |
| 2017, Steffen et al. [45] | Autologous canine BM-MSC <br> Canine, in vivo | Successful injection of BM-MSC into lumbosacral discs of naturally IVD-degenerative canines |
| 2017, Noriega et al. [46] | Allogeneic BM-MSC <br> Clinical trial (N = 24, follow-up: 12 months) | Significant VAS and ODI reductions, improvement on MRI |
| 2017, Centeno et al. [47] | Autologous BM-MSC <br> Clinical trial (N = 33, follow-up: 72 months) | Disc bulging reduction on MRI, pain and function improvement |
| 2017, Kumar et al. [48] | Autologous AD-MSC <br> Clinical trial (N = 10, follow-up: 12 months) | Combined implantation of AD-MSC and hyaluronic acid in discogenic back pain is safe and tolerable |
| 2017, Pettine et al. [49] | Autologous BM-MSC <br> Clinical trial (N = 26, follow-up: 36 months) | Evidence for the safety and feasibility of intradiscal BM concentrate therapy |
| 2016, Tschugg et al. [50] | Autologous disc chondrocyte <br> Clinical trial (N = 120, follow-up: 48 months) | Ongoing study |

BM-MSC, bone marrow-derived mesenchymal stem cell; AD-MSC, adipose tissue-derived mesenchymal stem cell; NP, nucleus pulposus; AF, annulus fibrosus; VAS, visual analogue scale; ODI, Oswestry disability index; iPSC, induced pluripotent stem cell; IVD, intervertebral disc.

To summarize, stem cell therapy can be used to induce IVD repair by preventing cellular apoptosis and inflammation, and by increasing the resident population and ECM production, and there is great interest in developing biomaterials for effective cell delivery, increasing cell viability, and inducing differentiation of stem cells into IVD-like cells.

## 3. Tissue Engineering for IVD Regeneration

Although many studies have reported alternative biological treatments for IVDD, these approaches have certain limitations. Direct administration of growth factors is associated with the short half-life of growth factors and potential lack of IVD cells as therapeutic targets in severe disc degeneration [29]. Gene therapy has several disadvantages, including inefficient gene delivery, unstable long-term expression, and lack of safety. While cell-based therapies have shown more promising therapeutic potential, the best strategies for effectiveness and safety have yet to be addressed [29]. In a clinical setting, stem cells are implanted in a harsh environment consisting of low cellularity, low glucose, low oxygen, low PH due to high lactic acid accumulation, low nutrients, and an inflammatory milieu [6,51–53]. Inflammatory mediators are a key component of progressive IVDD. All these factors can affect the differentiation potential, viability, and metabolism of the implanted stem cells [52]. MSCs can function optimally in inflammatory, hypoxic, acidic, and malnourished environments of the degenerated disc and have an immunomodulatory paracrine effect [52,54,55]. Hypoxia-exposed human MSCs (hMSCs) have been reported to improve tissue protection, but exposure of MSCs to inflammatory factors or hypoxic environment may adversely affect MSC differentiation. Therefore, it is important and necessary to design scaffolds

for effective cell delivery and induction of stem cell differentiation for tissue engineering applications (Figure 2).

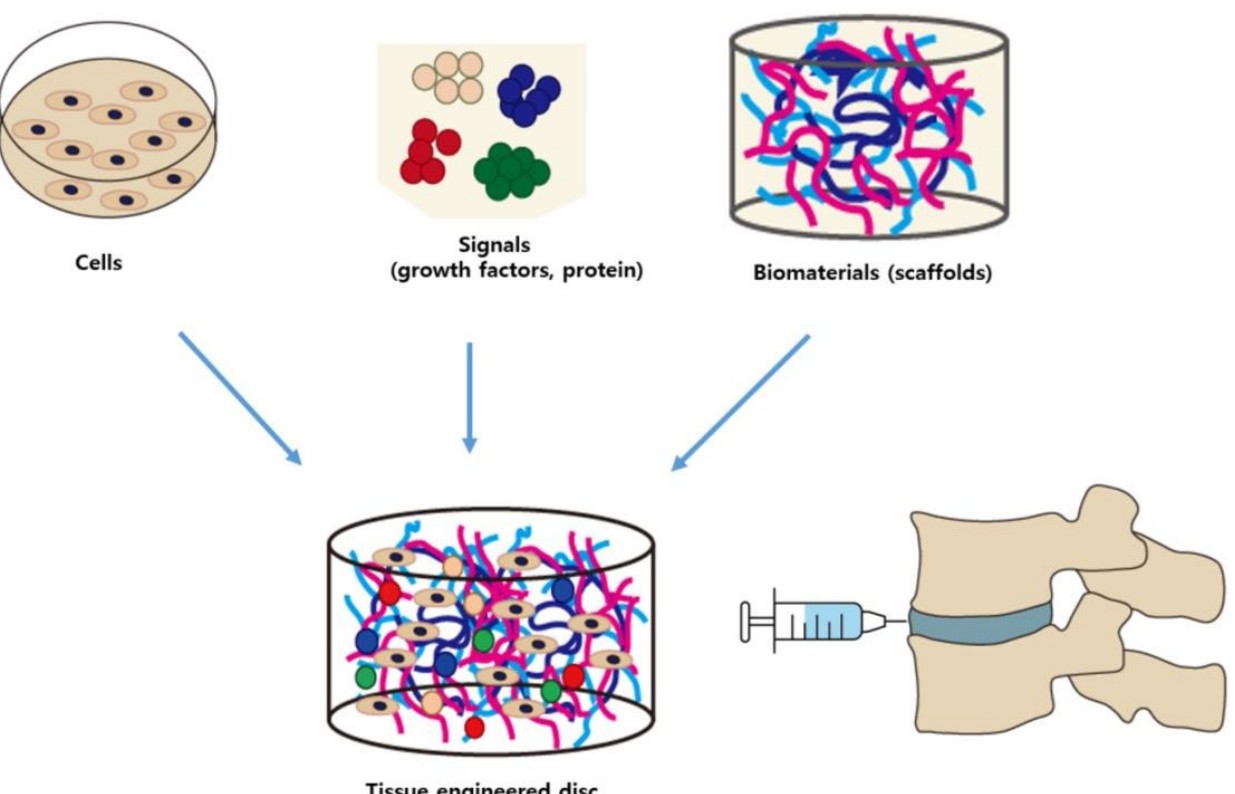

**Figure 2.** Combination of cells, signaling molecules (growth factors), and biomaterials for tissue engineering applications for intervertebral disc (IVD) degeneration.

### 3.1. Biomaterials

In the case of severe IVDD with loss of cell volume and of the physiological disc structure, the disc height must be restored to ensure IVD function [29]. This strategy involves a tissue engineering approach using biomaterials, which may serve as functional alternatives and scaffolds for the IVD tissue [13]. Researchers are using a composite approach that utilizes cell-loaded biomaterials to provide a structural environment for mechanical stability and potential cell regeneration [56]. The biomaterials used include injectable hydrogels and synthetic polymers designed from materials such as alginate, gelatin, polyglycolic acid, polylactic acid, hyaluronic acid (HA), and collagen [57].

Hydrogels can be used as an alternative to NP due to their biophysical properties and ability to absorb water, resist repeated loads, and act as a delivery vehicle [58]. An ideal injectable biomaterial will support cell retention and survival and make it possible to maintain or promote the NP phenotype in vivo. In the absence of biomaterials, cell injection leads to rapid cell death or migration from the injection site [59,60]. Important parameters to consider in the development of biomaterials are material viscosity, gelation rate, final gel stiffness, adhesivity, and degradation time controlled by polymer composition.

HA is a key component of the NP ECM that provides resistance to compression and allows for periodic loading [61,62]. Physiologically, HA has been demonstrated to have cartilage protective and anti-inflammatory properties, which have been shown to be associated with cell-based interventions. Therefore, some clinical trials have used HA as a cell carrier to increase the viscosity of the cell solution and enhance the retention of injected cells [63,64].

Collagen is one of the most widely used materials for tissue regeneration as it has numerous adhesion sites, limited immunogenicity, and is injectable. However, due to

its poor degradation and mechanical properties, it has not been widely used for disc repair. Composite collagen hydrogels, on the other hand, have been found to improve the compressive mechanical properties of the scaffold and control the rate of scaffold degradation [27].

Fibrin is a naturally occurring biomaterial that provides intrinsic physical and soluble cues to initiate tissue repair. Biodegradable fibrin hydrogels can be produced as injectable cell carriers and can be mediated by adjusting coagulation protein levels or altering the ionic strength of the system [65,66]. Fibrin-only hydrogels remain vulnerable to cell-mediated remodeling, while fibrin-HA composite hydrogels improve stability by increasing glycosaminoglycan (GAG) synthesis. In addition, silk to fibrin-HA gels significantly improve the mechanical properties and promote chondrogenesis [67]. Silk offers high resistance to compression, and silk-fiber stability, due to its wide range of hydrogen bonds, protein hydrophobicity, and high crystallinity, provides an advantage as a scaffold for IVD bioengineering [68].

Other biomaterials that can be used as a matrix to support AF and NP engineering are chitosan and alginate, which are inexpensive and easily accessible [12]. In addition, these two polymers have a synergistic effect combined with hybrid scaffolding [69]. Chitosan is used as a biodegradable and biocompatible polymer with low toxicity and excellent antibacterial properties. The soft, spongy chitosan-based scaffold has high porosity and pore interconnectivity to support cell adhesion and growth [70]. Alginate, one of the most abundant natural materials, mainly derived from brown algae and some bacteria, is used in a variety of biomedical applications and drug delivery systems due to its excellent biocompatibility, biodegradability, non-antigenicity, and chelation ability [69].

The use of synthetic materials as injectable fillers or cell carriers is a promising strategy to prevent the biomechanical limitations of natural polymer-based hydrogels. Many synthetic biomaterials, such as polyethylene glycol (PEG), PLGA, polyvinyl alcohol (PVA), polyvinylpyrrolidone (PVP), and hydroxyethyl methacrylate (HEMA), have been used as drug delivery and cell carriers [71,72]. The first approach used to restore NP height, function, and motion focused on the use of in situ hydrated synthetic polymers to restore NP hydration, IVD pressure, and disc height. This tactic aimed at mimicking the hydration properties of the NP glycosaminoglycans, which are slowly degraded and modified as they age and degenerate [57]. A copolymeric hydrogel, with the longest history of clinical use, has served as an alternative to NP. However, similar to previously used in situ hydrated polymers, some complications have already been reported, including gel fragmentation during swelling [73]. Other biomaterials, such as NP implant devices, have been developed from an injectable polymer that physically transitions to a gel or solid form. This approach has the advantage of minimizing tissue damage to AF during transplantation. In addition, various strategies, such as chemical cross-linking agents and heat- or pH-induced transition, have been developed. The cross-linked material inhibits proteolysis and induces the stiffness for disc implant [57].

Many scientists have attempted to develop monophasic NP or AF scaffolds such as ECM-based scaffold [74], adipose-derived MSCs (AD-MSCs)-loaded NP tissue-engineered construct [75], ASCs-seeded type II collagen/chondroitin sulfate composite hydrogel [76], and decellularized NP-based scaffold [77] (Table 2). Extensive efforts have also been made to develop biphasic tissue-engineered AF-NP composite scaffolds such as a collagen-GAG co-precipitate (NP-like core)/photochemically crosslinked collagen membranes (AF-like lamellae) composite [78], multiple AF-like lamellae encasing an NP-like core (MSCs-seeded collagen-GAG) [79], engineered nanofibrous disk-like angle-ply structure [80], biomimetic glycosaminoglycan analogues based on sulphonate-containing polymers [81], and multiple HA-PEG composite hydrogel [82] (Table 2). Animal studies using these biphasic scaffolds have shown excellent mechanical and biochemical results, suggesting that mimicking the morphology of the IVD is important for long-term stability and function of the implanted scaffolds [78–82].

**Table 2.** Biomaterials for IVD regeneration.

| Author | Materials | Effect |
|---|---|---|
| 2020, Penolazzi et al. [74] | Decellularized Wharton's jelly matrix from human umbilical cord as ECM-based scaffold | Promoted cell differentiation toward a discogenic phenotype, positively affected the expression of regulators of IVD homeostasis |
| 2019, Ishiguro et al. [75] | AD-MSC-Tissue engineered construct Rat | Regenerative efficacy was investigated structurally and biomechanically up to 6 months after implantation |
| 2018, Zhou et al. [76] | Type II collagen/chondroitin sulfate (CS) composite hydrogel-like adipose-derived stem cell delivery system | Minimally invasive approach to promote the regeneration of degenerated NP |
| 2018, Zhou et al. [77] | Injectable decellularized NP-based cell delivery system (NPCS) | The mechanical properties of the NPCS system were similar to those of fresh NP; Biocompatible; It induced NP-like differentiation and ECM synthesis |
| 2015, Choy et al. [78] | Collagen-glycosaminoglycan (GAG) co-precipitate and multiple lamellae of a photo-crosslinked collagen membrane | A biphasic scaffold comprising 10 AF-like lamellae had the best mechanical performance and elastic compliance |
| 2015, Chik et al. [79] | Collagen-GAG coprecipitate MSC and contracted collagen gel, MSC | Spinal motion segment tissue engineering. Provided a 3D model for studying tissue maturation and functional remodeling |
| 2014, Martin et al. [80] | Electrospun poly scaffold with cell-seeded hydrogels and disc-like angle-ply structure | Optimized the design of functional disc replacement in vivo |
| 2014, Sivan et al. [81] | Biomimetic GAG analogue based on sulphonate-containing polymer | Provided intrinsic swelling pressure which could maintain disc hydration and height |
| 2014, Jeong et al. [82] | Hyaluronic acid-poly(ethylene glycol) composite hydrogel | Highest number of NP and AF cells on HA-PEG hydrogels from lower molecular weight HA |

AD-MSC, adipose tissue-derived mesenchymal stem cell; GAG, glycosaminoglycan; HA, hyaluronic acid.

In addition, many studies have demonstrated that differentiation of stem cell into the IVD cell phenotype is promoted by various types of biomaterials; (1) natural biomaterials: collagen type II-chondroitin sulphate hydrogel, gelatin-HA methacrylate hydrogel, silk-protein-based multilayered angle-ply scaffold, chitosan-HA hydrogel, decellularized allogenic IVD, acellular porcine NP hydrogel, NP cell-derived acellular matrix, dextran chitosan and teleostean combined hydrogel, temperature sensitive hydrogel (chitosan-glycerophosphate), chitosan and alginate gel scaffold, alginate and chitosan hydrogels, and self-assembling peptides; (2) Synthetic biomaterials: poly(N-isopropylacrylamide (pNIPAM) hydrogel system, nanofibrous poly(l-lactide) (PLLA) scaffolds, and heparin-poly(ethylene argininylaspartate digylceride) (PEAD) conjugated vehicle; (3) Biosynthetic biomaterials: T1307-fibrinogen hydrogel, HA-pNIPAM hydrogel, and pentosan polysulfate (PEG-HA-PPS) hydrogel [83].

In summary, transplanted stem cells should survive, proliferate, and differentiate into NP-/AF-like cells. The combination of biomaterials and stem cells can provide an effective strategy to enhance effective cell delivery and stem cell differentiation capacity. Although a variety of biomaterials have been studied to investigate the effects of biomaterials on cell delivery and stem cell differentiation, few materials are currently available for clinical application due to the limitations of mechanical properties, immunogenicity, and uncontrollable deviations in inducing stem cells differentiation. In addition to the mechanical properties and biocompatibility of biomaterials, maintaining stem cells activity in a local niche and enhancing the ability of stem cells to differentiate into NP and AF cells facilitates the application of biomaterials in clinical practice [74–83].

### 3.2. Tissue Engineering for AF and NP Restoration and Maintenance

The aim of tissue engineering for IVD degeneration is the restoration and maintenance of both AF and NP anatomy and function. Tissue-engineered scaffolds must be able to withstand the physiological IVD loads and have excellent biocompatibility; proper porosity; and shapes, structures, and mechanical properties similar to those of IVD [79].

#### 3.2.1. AF Regeneration and Tissue Engineering

AF is composed of type I collagen and stacked lamellae and is highly organized [84]. AF is needed to transfer stress from the NP, maintain IVD integrity, and protect against damage caused by bending, stretching, and twisting [85,86]. The homeobox protein Mohawk (Mkx) has been reported to be essential for AF development, maintenance, and regeneration. It has been found that Mkx is predominantly expressed in the outer AF, and that removal of Mkx in mice resulted in the loss of numerous tendon- and ligament-related genes in the outer AF. Transplantation of MSCs overexpressing Mkx revealed the AF phenotype and promoted functional AF regeneration [87].

In AF regeneration and tissue engineering, natural materials, such as collagen, HA, chitosan, alginate, silk fibroin, and chondroitin sulfate, as well as natural biologic materials, such as the decellularized matrix from AF, are used to promote tissue regeneration and repair [57,88]. Natural scaffolds have the advantages of having low toxicity and similar properties to those of native tissue, and they can be mass-produced. Synthetic polymer scaffolds can be manufactured and processed based on the desired structural (aligned, angle-ply, hierarchical, bilayer, or biphasic) and mechanical properties of the final engineered tissue [11]. A poly(trimethylene carbonate) (PTMC) scaffold covered with a poly(ester-urethane) (PU) membrane to address AF rupture repair of bovine IVD has been manufactured as a carrier for MSCs. A PTMC scaffold with MSCs and PU membrane has been found to restore the disc height and prevent IVD herniation [89]. Furthermore, biodegradable poly(ether carbonate urethane)urea (PECUU) materials have been produced in AF-derived stem cells (AFSCs) using an electrospinning technique. Moreover, it has been reported that the elasticity of PECUU fibrous scaffolds with AFSCs resembled that of natural AF tissue [90].

#### 3.2.2. NP Regeneration and Tissue Engineering

NP is composed of type II collagen and PGs and contains 77% water. In recent years, bioengineered scaffolds that resemble the native NP structure and its mechanical properties have attracted attention [11]. Easy to inject high molecular weight hyaluronic acid-gelatin-adipic acid dehydrazide (oxi-HAG-ADH) hydrogels with anti-inflammatory and immunosuppressive activities, low viscosity, viscoelasticity similar to that of NP tissue, and expression of NP ECM genes have been fabricated [91]. Choi et al. generated hyaluronic-methylcellulose (HAMC) hydrogels loaded with Wharton's jelly-derived MSCs (WJ-MSCs), which significantly promoted degenerated disc repair by improving NP cell viability and decreasing ECM degradation [92]. Gan et al. generated a hydrogel with dextran and gelatin as the first network and PEG as the second network to produce hydrogels, forming the optimal 3D interpenetrating network hydrogel. This increased NP cell proliferation, long-term cell retention and survival, and promoted rehydration and regeneration of degenerative NP in animal models [93]. Laminin is the main component of the NP ECM and directly interacts with NP cells to regulate their function. Several laminin mimetic peptides bound to polyacrylamide gels have been reported to be able to support an immature and healthy NP phenotype. These hydrogel scaffolds provided a favorable environment for NP cell proliferation [94]. Wan et al. manufactured a biocompatible self-assembled peptide hydrogel (SAPH) with easily modifiable properties and nanofibrous architecture. They reported that the SAPH scaffold was as strong as native tissue, injectable, and that it restored the IVD cell phenotype and stimulated deposition of aggrecan and type II collagen, which are key NP ECM components [95].

### 3.2.3. NP-AF Regeneration and Tissue Engineering

NP-AF tissue engineering combines two approaches: NP replacement and AF repair. There are three ways to manufacture NP-AF, namely, using NP and AF cell-seeding scaffolds, integrated biphasic NP-AF scaffolds, and scaffolds made with decellularized natural IVD [11]. Scaffolds seeded with NP and AF cells were separately prepared and assembled into a composite construct. Nesti et al. manufactured a biphasic construct using electrospun MSCs seeded on a PLLA scaffold and HA hydrogel [96]. Choy et al. generated a biphasic NP-AF scaffold with integrated collagen and glycosaminoglycans. The biphasic scaffold was composed of collagen-glycosaminoglycan, which coprecipitates as an NP-like core, and encapsulated in multiple lamellae of photochemically cross-linked collagen membranes that made up AF-like lamellae [79]. This scaffold exhibited mechanical characteristics similar to those of native discs with a ring-independent height recovery of 82–89%. Park et al. generated a scaffold consisting of chondrocytes and AF cells, which were respectively seeded into a scaffold consisting of hydrogel in the center and silk protein in the periphery, respectively [97]. Yang et al. manufactured an IVD scaffold by inversely reconstructing the structure of native IVD and bioprinting bacterial cellulose nanofibers using a high-throughput-optimized micropattern screening microchip in rats [98]. Chan et al. made a 70% endogenous cell-removing scaffold that preserved the glycosaminoglycan content, collagen fibril structure, and mechanical properties of the IVD by altering chemical and physical decellularization [99]. Hensley et al. created a natural NP-AF composite scaffold using decellularized bovine tail IVD and confirmed the presence of type II collagen and glycosaminoglycan in the NP region and the native angle-ply collagen microarchitecture in the AF region [100].

### 3.2.4. Summary

Tissue engineering techniques have emerged as a possible approach to treat IVDD by replacing degenerated discs with appropriate stem cells and biomaterials. Tissue engineered AF and NP can restore their function by repairing or replacing degenerated discs. Therefore, considerable research is underway on the development of scaffolds suitable for AF and NP regeneration. Many natural and synthetic biomaterials can be used as supporting matrices in AF and NP scaffolds [12]. Advances in manufacturing technologies, material processing and development, surface functionalization, drug delivery systems, and cell integration have accelerated the development of tissue engineering therapies for IVDD.

## 4. Conclusions

Chronic LBP due to IVDD represents a significant health and social burden. Regenerative tactics are being investigated with significant advances in understanding the characteristics of IVDD (Table 3). Current promising strategies include growth factor injection, gene therapy, cell-based therapy, and tissue engineering using biomaterials. In this review, we investigated biological and tissue engineering approaches for the treatment of IVD degeneration and regeneration strategies. Limitations of biological approaches that remain to be overcome include the short half-life and possible lack of IVD endogenous cells associated with growth factor injection therapy, inefficient gene delivery, unstable long-term expression, and safety issues in gene therapy, and the inflammatory environment, low pH, low oxygen tension, and poor nutritional availability in cell-based therapies. Promising tissue engineering strategies using cells, growth factors, and biomaterials could be utilized to overcome these problems. With the development of tissue engineering, scaffolds are considered the 'holy grail' of IVD repair [11]. However, tissue engineering therapy remains challenging due to a lack of accurate understanding of the underlying molecular mechanisms and regulation of IVD physiology. To date, there are no FDA approved intradiscal therapies associated with tissue engineering therapy. Therefore, more sophisticated materials and strategies for clinical application need to be developed. In addition, accurate diagnosis of IVDD and evaluation of therapeutic effectiveness are critical to the develop-

ment of successful biological therapies. Although T2 mapping and diffusion weighted images(DWI) are newly quantified methods for IVDD evaluation [101], the development of improved non-destructive imaging techniques is essential to evaluate IVDD.

**Table 3.** Type of regenerative therapies for intervertebral disc regeneration.

| Type | | | Advantages | Disadvantages |
|---|---|---|---|---|
| Growth factor | | GDF-5, IGF-1, TGF-β, bFGF, OP-1 | Stimulation of ECM production | Short half-life<br>Need repeated injection |
| Gene therapy | | Virus mediated<br>Non-virus mediated<br>RNAi<br>CRISPR/Cas9 | Long-lasting and timeless effects | Safety concerns<br>Ethical concerns<br>Significant cost |
| Stem cell | ESCs | | Differentiation into three germ layers<br>Self-renewal and high replication | Immune rejection concern<br>Ethical concerns |
| | | | | Potential for tumor formation |
| | iPSCs | | Less ethical concerns than ESCs<br>Patient-specific<br>Autologous | Need method standardization<br>Potential for tumor formation<br>Need validation for safety |
| | MSCs | Bone marrow, Adipose tissue, umbilical cord Wharton's jelly Synovial membrane | Less ethical concerns than ESCs and iPSCs | Less cell proliferation<br>Limit differentiation potential |
| Tissue engineering | | Combination: stem cells, biomaterials, and growth factors | Ideal constructs | Need validation for biodegradation, biocompatibility, and optimal |

GDF-5, growth differentiation factor-5; IGF-1, insulin-like growth factor-1; TGF-β, transforming growth factor- β, bFGF, basic fibroblast growth factor, OP-1, osteogenic protein-1; ECM, extracellular matrix; RNAi, RNA interference; CRISPR/Cas9, Clustered Regulatory Interspaced Short Palindromic Repeats-Associated Cas9; ESCs, embryonic stem cells; iPSCs, induced pluripotent stem cells; MSCs, mesenchymal stem cells.

**Author Contributions:** Conceptualization and methodology: I.H.; writing: C.K.L., D.H.H. and I.H.; data acquisition: H.C. (Hungtae Chung), E.J.R., A.D., J.W.K., H.C. (Hyemin Choi), S.Y.K. and B.B. All authors have read and agreed to the published version of the manuscript.

**Funding:** This research was supported by the Korea Health Technology Research and Development Project, Ministry for Health and Welfare Affairs (HR16C0002, HI20C0579) and a grant of the National Research Foundation of Korea (NRF) (2020R1A2C4001870.

**Institutional Review Board Statement:** Not applicable.

**Informed Consent Statement:** Not applicable.

**Conflicts of Interest:** The authors declare no conflict of interest.

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
