# Peer review of "Advances in Tissue Engineering for Disc Repair"

_applsci, doi:10.3390/app11041919_

Round 1

Reviewer 1 Report

Therapeutic approaches for degeneration of IVD caused low back pain, thus to solve the LBP burden  the advance of tissue engineering is required.  Particularly, since the burden of can not be completely cured  conservative treatments, such as pain medication, anti inflammation medication, physical therapy, even with surgical treatment, a new and  creative approach is urgent to therapy of LBP. In this manuscript, authors offered an overview on the current advance  of tissue engineering approaches on  the treatment of IVD degeneration and summarized a serious development of new and creative therapies.  The manuscript is useful and interesting to the readers of this journal. However, there some minor weakness in this review.

 1, row 55- 58. “ Is eligible for surgical treatment. Surgical treatments include”,  a weird transition sentence. It sounds that surgery may solve the problems which conservative treatment, but actually, it is not.

2,  The isolation of mesenchymal stem cells  from such tissues , BM, adipose,  umbilical cord,  and synovium, are not just based on large numbers from a single donor. According literature by Dr. Caplan,  there are only 1,500 MSCs  of 10 million BM cells, it’s really a large number. There are lots of MSCs in muscle, but there a small number of approach using MSCs from muscle, since the injury of muscle is not easy healed and the surgery is painful.

3, Although authors mentioned the application of ES cells in cell based therapies, the application of iPS cells is the real hot topic in tissue engineering area.

4,  Author indicated the design of scaffolds  is important and necessary for cell delivery and proliferation. However, another function for inductive differentiation od scaffolds should be discussed in this part .

5, Row 276, “a lot of water”  is not accurate, it’s 77% percentage.

6, Authors have categorized the different approaches and  current advance in therapy,, however, the lack of summary for each part is difficult to understanding this article. I suggest that authors make a short summary at the end of each part.   

Author Response

I.1 Response to specific comments from Reviewer 1:

Overall Comment:

Therapeutic approaches for degeneration of IVD caused low back pain, thus to solve the LBP burden the advance of tissue engineering is required.  Particularly, since the burden of can not be completely cured  conservative treatments, such as pain medication, anti inflammation medication, physical therapy, even with surgical treatment, a new and  creative approach is urgent to therapy of LBP. In this manuscript, authors offered an overview on the current advance of tissue engineering approaches on the treatment of IVD degeneration and summarized a serious development of new and creative therapies.  The manuscript is useful and interesting to the readers of this journal. However, there some minor weakness in this review.

COMMENT 1:

row 55- 58. “Is eligible for surgical treatment. Surgical treatments include”, a weird transition sentence. It sounds that surgery may solve the problems which conservative treatment, but actually, it is not.

RESPONSE 1: We thank for the reviewer’s comment.

We modified the sentence as follows.

Line 54-55: Chronic LBP due to IVDD may be considered surgical treatments if there is no response to conservative treatments such as medication and physical therapy.

COMMENT 2: Line 133-134: The isolation of mesenchymal stem cells from such tissues, BM, adipose,  umbilical cord,  and synovium, are not just based on large numbers from a single donor. According literature by Dr. Caplan,  there are only 1,500 MSCs  of 10 million BM cells, it’s really a large number. There are lots of MSCs in muscle, but there a small number of approach using MSCs from muscle, since the injury of muscle is not easy healed and the surgery is painful.

RESPONSE 2: Thank you for pointing this out. We modified the sentence as follows.

(Line 163-164)

because these cells are relatively easy to obtain and can differentiate into chondrogenic and IVD-cell lineages

COMMENT 3: Although authors mentioned the application of ES cells in cell based therapies, the application of iPS cells is the real hot topic in tissue engineering area.

RESPONSE 3: We appreciate the reviewer’s concern. We added iPS cells into the cell based therapy section.

Notochordal cells have been suggested to cause disc degeneration because their loss is associated with the onset of IVDD[33]. Sheyn et al. demonstrated that notochordal-like cells from human induced pluripotent stem cells (iPSCs) reduce IVDD in an injury-induced porcine model. Therefore, the application of iPSCs is a hot topic in the field of IVD regeneration and has several advantages over embryonic stem cells (ESCs) such as ethics and immune rejection issues. However, iPSCs also have drawbacks in clinical applications such as tumor formation by genomic integration of reprogramming factors[36].

COMMENT 4: Author indicated the design of scaffolds is important and necessary for cell delivery and proliferation. However, another function for inductive differentiation of scaffolds should be discussed in this part.

RESPONSE 4: Thank you for pointing this out. We added this point.

Line 208-209: necessary to design scaffolds for effective cell delivery and induction of stem cell differentiation for tissue engineering applications

Line 271-280:

In addition, many studies have demonstrated that differentiation of stem cell into the IVD cell phenotype is promoted by various types of biomaterials; (1) natural biomaterials: collagen type II-chondroitin sulphate hydrogel, gelatin-HA methacrylate hydrogel, silk protein-based multilayered angle-ply scaffold, chitosan-HA hydrogel, decellularized allogenic IVD, acellular porcine NP hydrogel, NP cell-derived acellular matrix, Dextran chitosan & teleostean combined hydrogel, temperature sensitive hydrogel (chitosan-glycerophosphate), chitosan and alginate gel scaffold, alginate and chitosan hydrogels, and self-assembling peptides; (2) Synthetic biomaterials: pNIPAM (poly(N-isopropylacrylamide) hydrogel system, nanofibrous poly(l-lactide;PLLA) scaffolds, and heparin-PEAD conjugated vehicle; (3) Biosynthetic biomaterials: T1307-fibrinogen hydrogel, HA-pNIPAM hydrogel, and PEG-HA-PPS (pentosan polysulfate) hydrogel[74].

COMMENT 5: Row 276, “a lot of water”  is not accurate, it’s 77% percentage.

RESPONSE 5: Thank you for pointing this out. We modified the sentence.

Line 324: contains 77 % water.  

COMMENT 6: Authors have categorized the different approaches and  current advance in therapy,, however, the lack of summary for each part is difficult to understanding this article. I suggest that authors make a short summary at the end of each part.

RESPONSE 6: Thank you for pointing this out. a short summary at the end of each part.

Line 145-148: 2.1.3 Summary

In order to overcome the shortcomings of molecular therapy, future studies will focus on the delivery and controlled release to the degenerated discs. In addition, the combination of growth factors, stem cells and biomaterials should be a focus going forward.

Line 184-186: To summarize, stem cell therapy can be used to induce IVD repair by preventing cellular apoptosis and inflammation, and by increasing the resident population and ECM production, and there is great interest in developing biomaterials for effective cell delivery, increasing cell viability, and inducing differentiation of stem cells into IVD-like cells.

Line 84-392:

Tissue engineering techniques have emerged as a possible approach to treat IVDD by replacing degenerated discs with appropriate stem cells and biomaterials. Tissue engineered AF and NP can restore their function by repairing or replacing degenerated discs. Therefore, considerable research is underway on the development of scaffolds suitable for AF and NP regeneration. Many natural and synthetic biomaterials can be used as supporting matrices in AF and NP scaffolds[12]. Advances in manufacturing technologies, material processing and development, surface functionalization, drug delivery systems, and cell integration have accelerated the development of tissue engineering therapies for IVDD.

Reviewer 2 Report

The manuscript titled, “Advances in Tissue Engineering for Disc Repair” is a very important paper outlining the advancements in treatment of IVD degeneration. I have a few important points (see below).

Before section 2 the authors should briefly mention the changes and physical markers (i.e., any specific changes in physical/chemical characteristics of the disc, bone and spinal ligaments) and any biomarkers with time as this might determine what treatments might be more successful (mentioned in the following sections). It should be more emphasized in the introduction before the authors  mention at what stage of IVD degeneration the treatments mentioned in that section are most successful.

After looking at other reviews on the topic there are similar approaches for treatment in the manuscript and the authors should mention any other very new methods I have seen that are developed. For example the use of nanoparticles for the delivery of genes, growth factors and other agents for treatment.

Developments of the different imaging modalities for diagnosis should be briefly mentioned for the relevant sections as this significantly contributes to the success of treatments by being able to detect the physical/biological markers.

The inclusion of a table in the manuscript would be good for summarizing the advantages and disadvantages/limitations of the different methods.

Author Response

I.2 Response to specific comments from Reviewer 2:

Comment 1:

The manuscript titled, “Advances in Tissue Engineering for Disc Repair” is a very important paper outlining the advancements in treatment of IVD degeneration. I have a few important points (see below).

Before section 2 the authors should briefly mention the changes and physical markers (i.e., any specific changes in physical/chemical characteristics of the disc, bone and spinal ligaments) and any biomarkers with time as this might determine what treatments might be more successful (mentioned in the following sections). It should be more emphasized in the introduction before the authors  mention at what stage of IVD degeneration the treatments mentioned in that section are most successful.

RESPONSE 1.  Thank you for pointing this out. We added this point.

Line 78-93.

Depending on the stage of IVDD, different treatment strategies for managing IVDD have been recommended[14]. Initial IVDD shows change in the NP and AF matrices, while IVDD induces progressive structural changes such as annular fissures, disc herniation, disc height reduction, and disc space collapse. The Pfirrmann Disc Grading is a useful scoring tool for evaluating IVDD on MRI T2-weighted images; Grade I: homogeneous disc with bright high intensity and normal disc height, Grade II: inhomogeneous disc, but high intensity signal, clear distinction between NP and AF, and normal disc height, Grade III: inhomogeneous disc with an intermittent gray signal intensity, unclear distinction between NP and AF, normal or slightly reduced disc height, Grade IV: inhomogeneous disc with low intensity dark gray signal intensity. No more distinction between the NP and AF, slightly or moderately decreased disc height, Grade V: inhomogeneous disc with a hypointense black signal intensity, no more distinction between the nucleus and annulus, and collapsed disc space[15]. For severe nerve compression due to herniated disc and disc space collapse(Pfirrmann grades IV and V), surgical removal of herniated disc and/or spinal fusion surgery are required. However, patients with discogenic LBP due to Pfirrmann grade II and III IVDD may receive regenerative molecular therapies such as growth factors, genes, and cell therapy with or without biomaterials. In addition, tissue engineered NP and NF could be applied for patients with Pfirrmann grades IV and V.

Comment 2:

After looking at other reviews on the topic there are similar approaches for treatment in the manuscript and the authors should mention any other very new methods I have seen that are developed. For example the use of nanoparticles for the delivery of genes, growth factors and other agents for treatment.

RESPONSE 2.  Thank you for pointing this out. We added this point.

Line 118-122.

Thus, the right carrier is a matter to consider. In recent years, the use of biodegradable microspheres for controlled local drug delivery has become a valuable approach to overcome the drawbacks of growth factors. Yan et al. demonstrated that injection of GDF-5 loaded into poly(lactic-co-glycolic acid) (PLGA) microspheres could improve regenerative efficacy of GDF-5 in a rat model[28].

Comment 3:

Developments of the different imaging modalities for diagnosis should be briefly mentioned for the relevant sections as this significantly contributes to the success of treatments by being able to detect the physical/biological markers.

RESPONSE 3.  Thank you for pointing this out. We added this point.

Line 410-413

In addition, accurate diagnosis of IVDD and evaluation of therapeutic effectiveness are critical to the development of successful biological therapies. Although T2 mapping and diffusion weighted images(DWI) are newly quantified methods for IVDD evaluation[101], the development of improved non-destructive imaging techniques is essential to evaluate IVDD.

Comment 4:

The inclusion of a table in the manuscript would be good for summarizing the advantages and disadvantages/limitations of the different methods.

RESPONSE 4.  Thank you for pointing this out. We added Table. .

Table 3. Type of regenerative therapies for intervertebral disc regeneration

Type

Advantages

Disadvantages

Growth factor

GDF-5, IGF-1, TGF-β, bFGF, OP-1

Stimulation of ECM production

Short half-life

Need repeated injection

Gene therapy

Virus mediated

Non-virus mediated

RNAi

CRISPR/Cas9

Long-lasting and timeless effects

Safety concerns

Ethical concerns

Significant cost

Stem cell

ESCs

Differentiation into three germ layers

Self-renewal and high replication

Immune rejection concern

Ethical concerns

Potential for tumor formation

iPSCs

Less ethical concerns than ESCs

Patient-specific

Autologous

Need method standardization

Potential for tumor formation

Need validation for safety

MSCs

Bone marrow, Adipose tissue, umbilical cord Wharton’s jelly

Synovial membrane

Less ethical concerns than ESCs and iPSCs

Less cell proliferation

Limit differentiation potential

Tissue engineering

Combination: stem cells, biomaterials, and growth factors

Ideal constructs

Need validation for biodegradation, biocompatibility, and optimal

GDF-5, growth differentiation factor-5; IGF-1, insulin-like growth factor-1; TGF-β, transforming growth factor- β, bFGF, basic fibroblast growth factor, OP-1, osteogenic protein-1; ECM, extracellular matrix; RNAi, RNA interference; CRISPR/Cas9, Clustered Regulatory Interspaced Short Palindromic Repeats-Associated Cas9; ESCs, embryonic stem cells; iPSCs, induced pluripotent stem cells; MSCs, mesenchymal stem cells

Round 2

Reviewer 2 Report

The authors have addressed all my points and recommend publication.